# Internal Rotation Measurement of the Knee with Polymer-Based Capacitive Strain Gauges versus Mechanical Rotation Measurement Taking Gender Differences into Account: A Comparative Analysis

**DOI:** 10.3390/life14010142

**Published:** 2024-01-19

**Authors:** Hermann O. Mayr, Nikolaus Rosenstiel, Karthika S. Prakash, Laura Maria Comella, Peter Woias, Hagen Schmal, Michael Seidenstuecker

**Affiliations:** 1G.E.R.N. Tissue Replacement, Regeneration & Neogenesis, Department of Orthopedics and Trauma Surgery, Medical Center-Albert-Ludwigs-University of Freiburg, Faculty of Medicine, Albert-Ludwigs-University of Freiburg, Engesser Straße 4, 79108 Freiburg, Germany; hermann.mayr@uniklinik-freiburg.de (H.O.M.); nikolaus.rosenstiel@me.com (N.R.); 2Department of Orthopedics and Trauma Surgery, Medical Center Albert-Ludwigs-University of Freiburg, Faculty of Medicine, Albert-Ludwigs-University of Freiburg, Hugstetter Str. 55, 79106 Freiburg, Germany; hagen.schmal@uniklinik-freiburg.de; 3Kreiskrankenhaus Lörrach, Spitalstraße 25, 79539 Lörrach, Germany; 4Department of Microsystems Engineering, IMTEK Albert-Ludwigs-University of Freiburg, Geoges-Koehler-Allee 102, 79110 Freiburg, Germany; karthikasprakash@gmail.com (K.S.P.); laura.comella@imtek.uni-freiburg.de (L.M.C.); peter.woias@imtek.uni-freiburg.de (P.W.); 5Institute for Applied Research (IAF), Karlsruhe University of Applied Sciences (HKA), Moltkestraße 30, 76133 Karlsruhe, Germany

**Keywords:** knee rotation measurement, polymer-based capacitive strain gauges, validation, measurement instruments, knee laxity, Laxitester

## Abstract

With the conventional mechanical rotation measurement of joints, only static measurements are possible with the patient at rest. In the future, it would be interesting to carry out dynamic rotation measurements, for example, when walking or participating in sports. Therefore, a measurement method with an elastic polymer-based capacitive measuring system was developed and validated. In our system, the measurement setup was comprised of a capacitive strain gauge made from a polymer, which was connected to a flexible printed circuit board. The electronics integrated into the printed circuit board allowed data acquisition and transmission. As the sensor strip was elongated, it caused a change in the spacing between the strain gauge’s electrodes, leading to a modification in capacitance. Consequently, this alteration in capacitance enabled the measurement of strain. The measurement system was affixed to the knee by adhering the sensor to the skin in alignment with the anterolateral ligament (ALL), allowing the lower part of the sensor (made of silicone) and the circuit board to be in direct contact with the knee’s surface. It is important to note that the sensor should be attached without any prior stretching. To validate the system, an in vivo test was conducted on 10 healthy volunteers. The dorsiflexion of the ankle was set at 2 Nm using a torque meter to eliminate any rotational laxity in the ankle. A strain gauge sensor was affixed to the Gerdii’s tubercle along the course of the anterolateral ligament, just beneath the lateral epicondyle of the thigh. In three successive measurements, the internal rotation of the foot and, consequently, the lower leg was quantified with a 2 Nm torque. The alteration in the stretch mark’s length was then compared to the measured internal rotation angle using the static measuring device. A statistically significant difference between genders emerged in the internal rotation range of the knee (*p* = 0.003), with female participants displaying a greater range of rotation compared to their male counterparts. The polymer-based capacitive strain gauge exhibited consistent linearity across all measurements, remaining within the sensor’s initial 20% strain range. The comparison between length change and the knee’s internal rotation angle revealed a positive correlation (r = 1, *p* < 0.01). The current study shows that elastic polymer-based capacitive strain gauges are a reliable instrument for the internal rotation measurement of the knee. This will allow dynamic measurements in the future under many different settings. In addition, significant gender differences in the internal rotation angle were seen.

## 1. Introduction

In the last ten years, a rise in leisure sports activities such as jogging and running has paralleled a rise in knee injuries. Among these injuries, a prevalent one is the tearing of the anterior cruciate ligament (ACL), constituting 20% of all knee injuries [1]. In recent decades, the importance of devices and methods for measuring rotational knee laxity caused by ACL rupture has increased [2,3].

Currently, there are several examination methods for diagnosing ACL ruptures caused by internal rotation of the tibia, including physical examination and instrumental measurement methods. The differentiated diagnosis of the rotational laxity of the knee joint is the crucial prerequisite for the correct therapeutic approach [4]. To date, evidence of knee instability has mostly been based on subjective, examiner-dependent clinical tests. Therefore, occasionally, no diagnostic consensus is reached. Physical examination includes the Lachman test [5], the pivot shift test [6] and the anterior drawer test [7]. These methods measure the range of motion between the stationary (upper) and mobile (lower) knee to analyze the degree of laxity [8]. These physical examination methods are standardized in principle, but results often vary and depend on examiners [9,10].

Regrettably, there is a shortage of evidence-based treatment algorithms and consensus among orthopedic professionals concerning the rotational laxity observed in both initial and repeated ACL ruptures. A fundamental challenge lies in the complex and standardized determination of internal rotational instability. The clinical assessment involves tests such as the dial and pivot shift, which are subjective and can be affected by the positioning of the knee and examiner-induced motion. As a result, the reliability of interrater agreement in manual examination methods is constrained [11,12,13]. Many studies using measuring devices, therefore, attempt to imitate and objectify the pivot shift test. However, due to the complexity of this movement, this has, so far, only been possible to a limited extent. The commercially available arthrometers that have been frequently used to date are capable of anterior-posterior-translation to be measured objectively, but the rotation component is not taken into account. The sensitivity of these devices is not yet optimal. There is also a dependency of the measurement results on the positioning of the patient, the examiner’s dominant arm and involuntary muscle contractions [14].

With the Rolimeter [15], the manual pull cannot be objectified, which means that reproducibility suffers. The Telos device [16] often produces false negative measurements and cannot be used repeatedly postoperatively due to the high radiation exposure. Studies using electromagnetic sensors can reliably determine the rotation-dependent knee joint kinematics. However, the fixation systems of the lower leg are not always sufficient. According to clinical experience, the sometimes high torques and bending moments often lead to muscular counter-tension in the patient and are perceived as painful. Some of these measurements require general anesthesia [17]. Due to their complexity, they are rarely applicable in routine clinical diagnostics. Computer navigation [18] is only useful in connection with surgical procedures. It delivers reliable results but serves more to validate surgical techniques. Cadaver studies [19] are experimental laboratory studies. Amputates are used in which semi-active stabilizers, such as the iliotibial band and the semimembranosus muscle, have been severed. The biomechanical properties of the tissue are altered in the cadaver so that the transfer of the results to the in vivo situation is only possible to a limited extent. In some cases, high forces that are not clinically tolerated are applied to the knee joint. The force is often transmitted directly via the bone and, thus, creates conditions that cannot be reproduced in everyday clinical practice. Nevertheless, these studies are groundbreaking and important in fundamental research. Devices from the early days of rotation measurement, such as the “Rottometer” [20], show a large difference between suggested and real lower leg rotation. This bias is promoted by the low adhesion between the lower leg, ankle and foot on the one hand and the receiving fixation mechanism on the other. This problem is already partially taken into account in the “rigid” boot of Tsai’s working group [21]. However, when the ankle joint is in a neutral position, there is still considerable rotational laxity in this joint. Quinn et al. [22] already described, in 1991, for the axial examination in internal and external rotation, that significantly higher three-dimensional forces and torsional moments act on the knee joint when the foot and ankle are fixed. Some of the instrumented measurements require an elaborate set-up, including fluoroscopic [23,24] or magnetic resonance imaging [25,26]. Other torsion measurements apply the torque to the foot without locking the ankle, which can lead to a considerable overestimation of the rotational laxity [20,21].

The “Laxitester” device developed by our work group to measure rotational laxity [27] takes this fact into account by fixing the talus in a defined dorsiflexion. The advantage of the device is the possibility of a controlled measurement of the lower leg rotation with a defined torque. It is also a purely mechanical, easily transportable device. But the Laxitester is a static measuring device. These measurements are only possible when the patient is at rest. All the instruments for measuring knee laxity are strongly influenced by the experience of the physician performing the measurement [9,10,28]. In the future, it would be interesting to perform dynamic rotational measurements, for example, during walking or sports. Therefore, a measurement method with a polymer-based sensor elastic–capacitive measurement system was developed and validated.

Hypothesis of the working group: reliable dynamic rotational measurements of the knee joint with polymer-based sensors are possible. There are significant gender differences in the rotational laxity of the knee joint.

## 2. Materials and Methods

A comparative analysis of the internal rotation measurement of the knee joint was carried out at the University Hospital of Freiburg from February to June 2022. The study was approved by the Ethics Committee of Freiburg University on 7 December 2021 (approval number 487/16).

### 2.1. Materials

The polymer-based sensor was fabricated using carbon black polydimethylsiloxane (C-PDMS) on the sensing part and pure PDMS as the substrate. The substrate was composed of PDMS, which was derived from a combination of two primary materials. This blend involved the amalgamation of two silicones, Neukasil^®®^ RTV-23 and RTV-17 (both sourced from Altropol Kunststoff GmbH, Stockelsdorf, Germany), in a weight ratio of 10:4. The resulting mixture was degassed to eliminate all entrapped air bubbles. For the conductive layer, an electrically conductive PDMS blend, referred to as C-PDMS (carbon black PDMS), was formulated by further integrating this silicone blend with 10.6 wt% of carbon black powder ENSACO^®®^ 250 P (TIMCAL Ltd., Bodio, Switzerland).

### 2.2. Methods

The production of the polymer-based capacitive strain gauge, as well as the printed circuit board, was carried out as already described by us [29]. The C-PDMS and PDMS were mixed together using a blade mixer for 10 min at 1200 rpm. The resulting mixture had a high viscosity, leading to uneven distribution in the mold. To address this issue, the viscosity of the mixture was reduced by adding 2.04 g of n-heptane. In the fabrication of the sensor, Neukasil silicone was employed. To establish electrical contact, a mushroom-shaped aluminum pin was embedded. Molds featuring holes were utilized during the fabrication process, consisting of ten holes with a diameter of 1.05 mm and a depth of 1.40 mm. These holes were designed to accommodate rigid electrical contacts when molding the C-PDMS layer. These rigid electrical contacts were precision-manufactured from aluminum through a CNC machine. They possessed a mushroom-like structure with small holes at their apex to ensure secure mechanical fixation and electrical connectivity when embedded in the C-PDMS layer. To maintain a stable connection with the C-PDMS, even during sensor extension, the contact pads were coated with a primer (NuSil SP-120, Silicon Primer, Songhan Plastic Technology Co., Ltd., Shanghai, China). The slender base of the contact was positioned in the mold’s holes, while the crown extended into the mold cavity, becoming completely enveloped by C-PDMS from all sides and through its openings. The measurement setup was comprised of a capacitive strain gauge based on the polymer linked to a flexible printed circuit board. The electronic components on this circuit board were employed for both data acquisition and transmission. When the sensor strip is elongated, it results in an alteration in the spacing between the strain gauge’s components, consequently leading to changes in capacitance. As previously explained, this change in capacitance allows for the measurement of strain [29]. The measurement system was affixed to the knee by adhering the sensor onto the skin in alignment with the anterolateral ligament (ALL). The lower part of the sensor, made of silicone, and the base of the circuit board made contact with the knee’s surface. Importantly, the sensor was applied to the knee without any prior stretching, as depicted in Figure 1.

The system was validated with an in vivo test on 10 subjects (5 females/5 males). For this purpose, the internal rotational laxity of the knee was tested on healthy volunteers, and the results were compared with the measurements on a static measuring device (Laxitester). The participants were positioned in a supine posture. The clinical examination commenced with the left leg and subsequently proceeded to the right leg, with each examination being meticulously documented using a designated form. Exclusion criteria included the presence of any abnormal findings in the ligaments in the collateral or cruciate ligament area, a history of prior knee surgeries or any restrictions in the range of motion. The lower part of the thigh was supported on a bench adjusted to achieve a knee joint flexion of 30°. To prevent rotation and displacement of the thigh, the femoral condyles were securely held in place by posts within a positioning aid. Additionally, this setup effectively prevented internal rotation of the thigh. The foot was secured in a retention device without the use of footwear (as depicted in Figure 2a). To ensure precise positioning, the second toe’s alignment was fine-tuned using an adjustable side clamp situated at the foot’s medial and lateral edges. Dorsiflexion was set at 2 Nm with a torque meter, in accordance with previous studies [27], to rule out rotational laxity in the ankle. The strain gauge sensor, based on polymer, was affixed to the Gerdii’s tubercle along the path of the anterolateral ligament, just beneath the lateral epicondyle of the thigh. Subsequently, the internal rotation of the foot and, consequently, the lower leg was measured in three successive assessments using a 2 Nm torque. The alteration in the stretch mark’s length was then juxtaposed with the measured internal rotation angle using the static measuring device Laxitester, as illustrated in Figure 2b.

The actual measurement was performed stepwise at 5°, with at least 20 s spent in the position after each step before the next 5° step was taken until the maximum knee rotation of the respective subject was reached. Each test was repeated three times. Prior to the measurements, a multiple (3–5) run-through of the measurement procedure was performed to allow the subjects to become accustomed to it and to relax them. Using calibration curves as already described in a previous work [29,30], the conversion of the “clock-ticks” into length change [31] was carried out. The change in capacitance of the sensor (due to the change in length) was measured as the time required to discharge the capacitance; this is called clock–tick [29].

### 2.3. Statistics

Descriptive statistics, linear regression and Spearman’s rank correlation were conducted in this study. The statistical analysis was carried out using SPSS software version 12.0 (SPSS Inc., Chicago, IL, USA). To describe the patient population, basic descriptive statistics were employed. Differences between groups were assessed with respect to age and sex, while the changes in measured length were compared among participants at varying rotation angles with increments of 5° (i.e., 10°, 15°, 20°, 25°, 30°, 35°, 40°). Scatter plots were utilized to assess disparities between participants’ values and variations across different angles.

For all participants, the correlation between the change in length and the degree of knee rotation was calculated using Spearman’s rank correlation coefficient (r) and significance level (*p*). The collected correlation data for different angles were then graphically represented. The significance level was set at 0.05.

Due to the ethics committee vote, only members of the working group were allowed to participate during the COVID-19 pandemic. Therefore, unfortunately, a power calculation was not possible.

## 3. Results

The current study included a total of 10 subjects, of which five were female and five were male. The average age was 28 ± 6 years.

The measurements showed a highly significant correlation between the purely mechanical measurement under maximum force and the digital measurement during dynamic examination with strain gauges (Figure 3). Furthermore, a significant difference in laxity was found between males and females (Table 1). Each test was repeated three times. The maximum standard deviation per individual was ±0.012 after circumference correction of length change.

According to Table 1, there are clear differences between the genders. The internal knee rotation of the females was up to 45° in some cases, whereas for most of the males, only 25° was possible. In addition, different values for the change in length were found for the same angle of rotation. The reason for this was the different sizes of the test subjects. When analyzing the data, it became apparent that it was necessary to mathematically correct the changes in length recorded using the polymer-based capacitive strain gauges depending on the leg circumference of the person examined (Figure 4). This means that in order to convert the change in length to an angle measurement, a correction factor must be taken into account. In order to compensate for the change in the length of the strain gauge with the same internal rotation, which is related to the size of the test person (and is also caused by the knee circumference), the following correction factor was used: one divided by the knee circumference of the test person.

Table 1 additionally shows that there is a significant difference in the length change between males and females with the same internal rotation angles. However, this was not exclusive to females as there were also males who demonstrated a significantly smaller length change (0.052 mm vs. 0.499 mm). If the correction factor (one divided by the knee circumference of the test person) is taken into account, the values become relative, and, in addition, a dimensionless knee circumference-corrected length change is obtained. However, it is still noticeable (Figure 4) that there are individual differences. These are due to the thickness of the tissue, the amount of muscle and the elasticity of the skin.

## 4. Discussion

The study showed a highly significant correlation between the purely mechanical measurement under maximum force and the digital measurement during dynamic examination with strain gauges. Furthermore, a significant difference in laxity was found between males and females.

The role of rotational laxity and, subsequently, extraarticular stabilizers was already noted in 1989 at a meeting of the American Orthopaedic Society for Sports Medicine (AOSSM) at an expert conference [32]. In a current review, Garcia-Mansilla et al. noted the importance of the anterolateral complex [25]. Especially in the case of recurrent anterior cruciate ligament ruptures, excessive rotational laxity is evident [33]. Studies increasingly support the importance of the anterolateral ligament in anterior cruciate ligament reconstruction [31].

A few processes have been established to structure elastic and highly flexible materials, such as polydimethylsiloxane (PDMS). Some of them allow one to combine conductive and nonconductive PDMS in one structure to construct flexible sensors [4]. This non-invasive measurement PDMS strain sensor is applied to the skin in the course of the anterolateral ligament. Based on the in vitro measurements of the simplified knee joint model of Zens et al. [34], the results obtained from the knee simulator exhibited a reproducible relative capacitance change of 0.01 for every 5° angle change from 10° through 45°. Already, in this in vitro experimental setup, it could be shown that the bone-to-bone displacement is transferred to the skin and measured there with the strain sensors. This could also be performed on human cadavers with capacitive strain sensors. As part of the project, an optimized polymer-based capacitive strain gauge was developed (Figure 1c). The sensor consists of two layers: a stretchable substrate and a stretchable conductive layer sitting on top of that, which is structured as a capacitive electrode. The sensor was first characterized. For this purpose, the sensor was subjected to a stretch of 10 mm and a relaxation for several cycles. The speed at which the sensor was stretched from one step to the next was 3 mm/sec. The precise stretching of the sensor was achieved by using a linear stage. The results confirmed the reproducibility of the sensors and the reliability of the newly developed manufacturing process. The differentiated diagnosis of rotational laxity of the knee joint is the crucial prerequisite for the correct therapeutic approach [31].

The Laxitester can only be used to measure the dorsiflexion when applying the maximum torque (2 Nm)—other statements regarding the kinetics of the knee cannot be made. The advantage of a digital measurement using polymer-based capacitive strain gauges compared to a mechanical internal rotation measurement with a fixed internal rotation force is that a linear measurement of internal rotation laxity is possible without applying a maximum force [35]. In addition, using our sensor, the measurements were reproducible in independent studies [36] and, compared to the Laxitester, our approach is independent of the investigator, which is one of the main points of criticism in using the Laxitester [9,10,28]. Using our sensor, the only dependent variable is the size of the subject, which could be corrected by a correction factor (based on the knee circumference). Nevertheless, individual patient-based differences can occur. The advantage of the method described here is that it is easy to use in everyday clinical practice—no time is required at the MRI scanner as is the case described in Kernkamp et al. [37] and Kosy et al. [38]. Moreover, the measurements can easily be performed directly on the patient and do not require any cadaveric studies, as in Lording et al. [39] or Wang et al. [40]. In addition, there is no need for camera or sensor systems such as in electromagnetic tracking, which the subjects walk past or in which the subjects perform movements [41]. An additional benefit of our system is that, after calibration has been performed for each patient, the strain gauges can also be used for other dynamic measurements such as walking, running or stair climbing. The data could then be sent directly to the patient’s or athlete’s smartphone via Bluetooth, for example, in order to record these measurements during everyday activities or directly during sports.

## 5. Limitations

The low number of cases due to the COVID-19 pandemic and ethical regulations is a major weakness. Patients with cruciate ligament pathologies were not included due to the ethical regulations.

## 6. Conclusions

The current study shows that elastic polymer-based capacitive strain gauges are a reliable instrument for internal rotation measurement of the knee. This will allow dynamic measurements in the future in many different settings. In addition, significant gender differences in the internal rotation angle were seen.

## Figures and Tables

**Figure 1 life-14-00142-f001:**
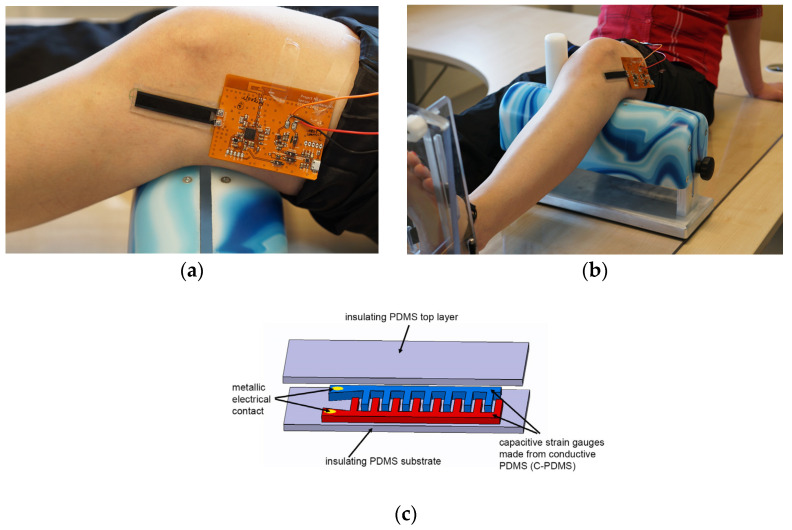
Images showing (**a**,**b**) positioning of the sensor, including the polymer-based capacitive strain gauges on the knee during measurements; (**c**) schematic sketch of the sensor. Note that the circuit board contained a thin rubber mat to provide insulation from below.

**Figure 2 life-14-00142-f002:**
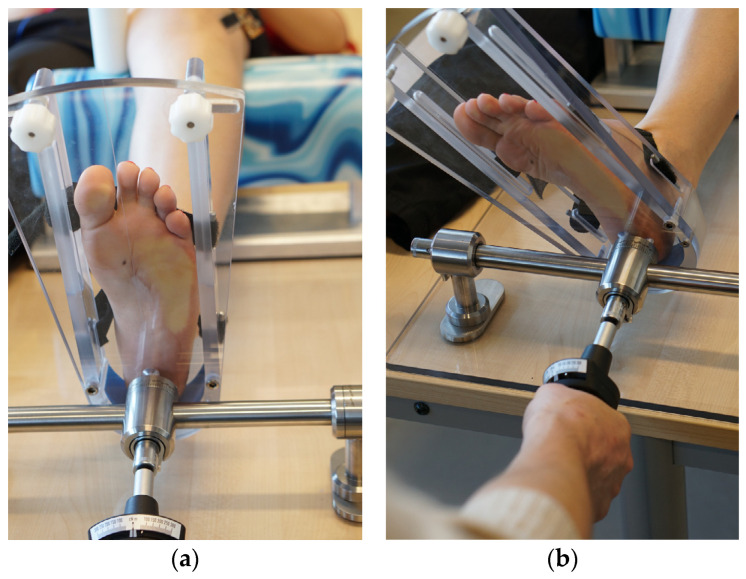
Measurement of knee laxity using the Laxitester static measuring device, he foot was secured in a retention device without the use of footwear, neutral position (**a**); measuring was carried out with a torque measurement up to a maximum of 2 Nm (**b**).

**Figure 3 life-14-00142-f003:**
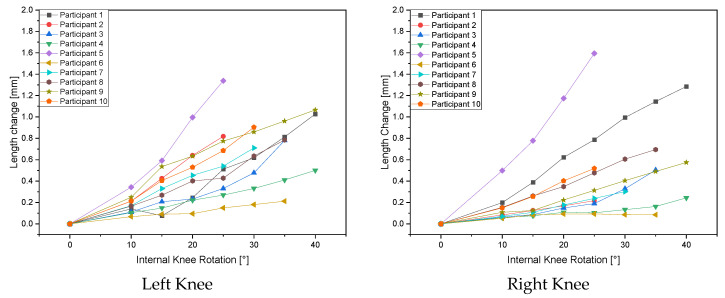
Comparison of the length change of the polymer-based capacitive strain gauge as a function of the internal knee rotation for all participants.

**Figure 4 life-14-00142-f004:**
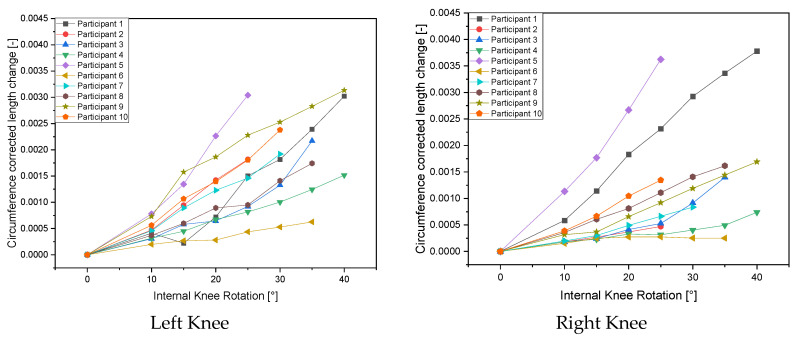
Circumference-corrected length change as a function of internal knee rotation.

**Table 1 life-14-00142-t001:** Length change of the polymer-based capacitive strain gauge as a function of knee internal rotation comparing the left and right knee of females vs. males.

	Length Change in mm [31]							
Internal Knee Rotation	0°	10°	15°	20°	25°	30°	35°	40°
	Left Knee Females							
Female 1	0	0.140	0.076	0.245	0.512	0.618	0.813	1.028
Female 2	0	0.110	0.208	0.232	0.331	0.478	0.781	
Female 3	0	0.104	0.148	0.222	0.270	0.332	0.410	0.500
Female 4	0	0.166	0.269	0.402	0.428	0.635	0.784	
Female 5	0	0.249	0.536	0.634	0.775	0.859	0.961	1.066
	Left Knee Males							
Male 1	0	0.212	0.426	0.640	0.818			
Male 2	0	0.344	0.592	0.996	1.338			
Male 3	0	0.068	0.091	0.095	0.149	0.180	0.213	
Male 4	0	0.170	0.330	0.454	0.540	0.710		
Male 5	0	0.213	0.405	0.530	0.686	0.904		
	Right Knee Females							
Female 1	0	0.199	0.388	0.623	0.787	0.994	1.144	1.285
Female 2	0	0.064	0.087	0.149	0.190	0.329	0.503	
Female 3	0	0.059	0.076	0.108	0.104	0.133	0.163	0.244
Female 4	0	0.154	0.261	0.349	0.477	0.605	0.695	
Female 5	0	0.108	0.126	0.223	0.313	0.404	0.490	0.575
	Right Knee Males							
Male 1	0	0.084	0.124	0.169	0.217			
Male 2	0	0.499	0.778	1.175	1.594			
Male 3	0	0.052	0.084	0.093	0.093	0.086	0.085	
Male 4	0	0.071	0.109	0.177	0.239	0.301		
Male 5	0	0.149	0.256	0.402	0.518			

## Data Availability

The data presented in this study are available upon request from the corresponding author.

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
