# Peer review of "Internal Rotation Measurement of the Knee with Polymer-Based Capacitive Strain Gauges versus Mechanical Rotation Measurement Taking Gender Differences into Account: A Comparative Analysis"

_life, 2024, doi:10.3390/life14010142_

Round 1

Reviewer 1 Report

Comments and Suggestions for Authors

Authors performed an study regarding the internal rotation measurement of the knee joint with polymer-based capacitive strain gauges compared to conventional mechanical rotation measurement. 

In my opinion, authors have written an appropiate introduction for the context of their research. However, I would recommend them to clarify their objective and their hypothesis. 

The methods are clearly described, although the information about the characteristics of the participants, inclusion and exclusion criteria, informed consent, etc; should be more explicit on the manuscript. Also, the statistical analysis does not match with the title neither the objective of the manuscript. Please, adapt your objectives and your title in order to englobe all your objectives, not only the comparison between both methods of measurement. 

Then the results are clearly described and the graphics help understand these results, and same with the discussion. 

Finally, I would recommend to change the conclusion to match the objectives, once you modify them. 

Author Response

Dear Reviewer 1,

thank you very much for taking the time to review our manuscript.

Also, the statistical analysis does not match with the title neither the objective of the manuscript. Please, adapt your objectives and your title in order to englobe all your objectives, not only the comparison between both methods of measurement.

Answer of the authors:

The title is now adapted to statistical analysis objectives:

Internal rotation measurement of the knee with polymer-based capacitive strain gauges versus mechanical rotation measurement taking gender differences into account a comparative analysis

I would recommend to change the conclusion to match the objectives

Answer of the authors:

Conclusion is now improved:

The current study shows that elastic polymer-based capacitive strain gauges are a reliable instrument for internal rotation measurement of the knee. This will allow dynamic measurements in the future under many different settings.  In addition, significant gender differences in the internal rotation angle were seen.

best regards and happy new year! 

the authors

Reviewer 2 Report

Comments and Suggestions for Authors

This study entitled “Internal rotation measurement of the knee joint with polymer-based capacitive strain gauges compared to conventional mechanical rotation measurement” seems to have been generally well executed and written. Furthermore, I believe that this paper will be of great interest to the readers. However, I have only remarks that require authors attention and few suggestions to further improve the quality of the paper. 

Title

Please add the type of article in your title.

Keywords

Consider some additional MeSH keywords to readers easier identify your research.

1.Introduction

Please make paragraph throughout Introduction to improve the readability.

Please state the clear hypothesis of your study at the end of Introduction.

2.Materials and Methods

Please begin this section with an information what type of study you have performed, in which time period and where. Following that include the Ethical approval for conducting your study (the number of approval and the date when the approval was gained). If you registered your study (e.g., ClinicalTrials.gov) please include the number and the date of registration.

Please make the subsections to improve the readability.

2.3 Statistics

Why the sample size calculation was not performed?

3. Results

Please state in the Results how many the participants were men and how many were women. Furthermore, state the mean or median age of your investigated population.

4.Discussion

Please begin Discussion with the main findings of your study.

Please expand the Discussion with the relevant results of similar studies.

Please state the limitations of your study at the end of Discussion.

Author Response

Dear Reviewer 2,

thank you very much for taking the time to review our manuscript. Our answers are highlighted in blue.

Title

Please add the type of article in your title.

We added the article type.

Keywords

Consider some additional MeSH keywords to readers easier identify your research.

Keywords are now improved:

Keywords: knee rotation measurement; polymer-based capacitive strain gauges; validation; measurement instruments; knee laxity; Laxitester

1.Introduction

Please make paragraph throughout Introduction to improve the readability.

Please state the clear hypothesis of your study at the end of Introduction.

Paragraph throughout Introduction is now made.

Clear hypothesis of your study has now been formulated

2.Materials and Methods

Please begin this section with an information what type of study you have performed, in which time period and where. Following that include the Ethical approval for conducting your study (the number of approval and the date when the approval was gained). If you registered your study (e.g., ClinicalTrials.gov) please include the number and the date of registration.

Please make the subsections to improve the readability.

Materials and Methods are now improved according to the reviewer's instructions.

2.3 Statistics

Why the sample size calculation was not performed?

We added the following to the text: 

Due to the ethics committee vote, only members of the working group were allowed to participate during the Covid pandemic. Therefore, unfortunately a power calculation was not possible.

3. Results

Please state in the Results how many the participants were men and how many were women. Furthermore, state the mean or median age of your investigated population.

Results are now completed.

4.Discussion

Please begin Discussion with the main findings of your study.

Please expand the Discussion with the relevant results of similar studies.

Please state the limitations of your study at the end of Discussion.

Discussion is now improved according to the reviewer's instructions. Pls see Lines 338-340

Best regards and a happy new year

the authors